# Epidemiology of Candidemia and Fluconazole Resistance in an ICU before and during the COVID-19 Pandemic Era

**DOI:** 10.3390/antibiotics11060771

**Published:** 2022-06-04

**Authors:** Christina Routsi, Joseph Meletiadis, Efstratia Charitidou, Aikaterini Gkoufa, Stelios Kokkoris, Stavros Karageorgiou, Charalampos Giannopoulos, Despoina Koulenti, Petros Andrikogiannopoulos, Efstathia Perivolioti, Athina Argyropoulou, Ioannis Vasileiadis, Georgia Vrioni, Elizabeth Paramythiotou

**Affiliations:** 1First Department of Intensive Care, School of Medicine, National and Kapodistrian University of Athens, Evangelismos Hospital, 10676 Athens, Greece; ef.charitidou@gmail.com (E.C.); katergouf@yahoo.gr (A.G.); skokkoris2003@yahoo.gr (S.K.); stavros99k@gmail.com (S.K.); harry.giannopoulos@gmail.com (C.G.); ioannisvmed@yahoo.gr (I.V.); 2Clinical Microbiology Laboratory, School of Medicine, National and Kapodistrian University of Athens, Attikon University Hospital, 12461 Athens, Greece; jmeletiadis@med.uoa.gr; 3Second Department of Intensive Care, School of Medicine, National and Kapodistrian University of Athens, Attikon University Hospital, 12461 Athens, Greece; d.koulenti@uq.edu.au (D.K.); lparamyth61@hotmail.com (E.P.); 4Department of Clinical Microbiology, Evangelismos Hospital, 10676 Athens, Greece; velavalton@hotmail.gr (P.A.); perivolioti@yahoo.gr (E.P.); athina.argyropoulou@gmail.com (A.A.); 5Department of Microbiology, School of Medicine, National and Kapodistrian University of Athens, 11527 Athens, Greece; gvrioni@med.uoa.gr

**Keywords:** candidemia, ICU, incidence, epidemiology, *Candida* species, non-*albicans Candida* species, fluconazole resistance, COVID-19, critically ill

## Abstract

The objectives of this study were to investigate the incidence of candidemia, as well as the factors associated with *Candida* species distribution and fluconazole resistance, among patients admitted to the intensive care unit (ICU) during the COVID-19 pandemic, as compared to two pre-pandemic periods. All patients admitted to the ICU due to COVID-19 from March 2020 to October 2021, as well as during two pre-pandemic periods (2005–2008 and 2012–2015), who developed candidemia, were included. During the COVID-19 study period, the incidence of candidemia was 10.2%, significantly higher compared with 3.2% and 4.2% in the two pre-pandemic periods, respectively. The proportion of non-*albicans Candida* species increased (from 60.6% to 62.3% and 75.8%, respectively), with a predominance of *C. parapsilosis.* A marked increase in fluconazole resistance (from 31% to 37.7% and 48.4%, respectively) was also observed. Regarding the total patient population with candidemia (*n* = 205), fluconazole resistance was independently associated with ICU length of stay (LOS) before candidemia (OR 1.03; CI: 1.01–1.06, *p* = 0.003), whereas the presence of shock at candidemia onset was associated with *C. albicans* (OR 6.89; CI: 2.2–25, *p* = 0.001), and with fluconazole-susceptible species (OR 0.23; CI: 0.07–0.64, *p* = 0.006). In conclusion, substantial increases in the incidence of candidemia, in non-*albicans*
*Candida* species, and in fluconazole resistance were found in patients admitted to the ICU due to COVID-19, compared to pre-pandemic periods. At candidemia onset, prolonged ICU LOS was associated with fluconazole-resistant and the presence of shock with fluconazole-susceptible species.

## 1. Introduction

Coronavirus disease 2019 (COVID-19), caused by the severe acute respiratory syndrome coronavirus 2 (SARS-CoV-2), being declared a pandemic by the World Health Organization on 11 March 2020 [1], spread rapidly around the world, causing a global health emergency [2]. Severe forms are complicated by hypoxemic acute respiratory failure requiring intensive care unit (ICU) admission [3,4]. In these patients, secondary infections, both bacterial and fungal, have been increasingly reported [5,6,7,8,9], resulting in the widespread use of antibiotics for the empirical treatment of suspected as well as of microbiologically confirmed infections, hence contributing to an increase in multidrug-resistant bacteria and fungi and increased costs of care [10].

Regarding fungal infections, a growing number of studies have mainly focused on *Aspergillus* superinfections in mechanically ventilated patients admitted to the ICU due to COVID-19, whereas bloodstream infections due to *Candida* species have been less studied thus far [11,12,13,14,15,16]. On the other hand, candidemia’s incidence, often cited as the fourth most common bloodstream infection in the ICU [17], is increasing, particularly in ICU patients [18,19]. In addition, the epidemiology of candidemia may change over time and can vary significantly across different geographic regions and hospitals. Furthermore, emerging azole resistance displays major challenges for therapeutic strategies [20,21]. Information on the epidemiology of candidemia in the ICU remains limited in the context of the ongoing COVID-19 pandemic. The objectives of the present study were to investigate the incidence of candidemia, as well as the factors associated with *Candida* species distribution and fluconazole resistance, among patients admitted to the intensive care unit (ICU) during the COVID-19 pandemic, as compared to two pre-pandemic periods.

## 2. Methods

### 2.1. Study Setting and Design

All patients with SARS-CoV-2 infection confirmed by reverse transcription polymerase chain reaction on nasopharyngeal swabs, and acute respiratory failure, admitted to the COVID-19 ICUs of ‘Evangelismos’ Hospital, a tertiary-care medical center, from March 2020 to October 2021, who developed candidemia during their ICU stay, constituted the COVID-19 candidemia cohort. Candidemia cases were identified through the electronic system. Approval for the use of the de-identified data was obtained from the ethics committee of the hospital (approval number 116/03-2021). Demographics, dates of hospital and ICU admissions, date of candidemia, detected *Candida* species, admission diagnosis classified as medical or surgical, main co-morbidities including diabetes mellitus and current malignancy, illness severity, length of stay (LOS) in ICU and ICU clinical outcome were recorded. The severity of acute illness was evaluated by the Acute Physiology and Chronic Health Evaluation (APACHE) II score [22] on ICU admission. The severity of organ dysfunction was assessed by the Sequential Organ Failure Assessment (SOFA) score [23], calculated on the first day of ICU admission and, additionally, on the day of candidemia. The difference (Delta) in the SOFA score, defined as the SOFA score on the ICU day that the positive blood culture for *Candida* species was collected minus the SOFA score on ICU admission, was also calculated. For the management and therapy of the ICU patients with COVID-19, international recommendations were followed [24]. For the treatment of candidemia, recommendations for application in non-immunocompromised critically ill patients were followed [25]. Accordingly, after candidemia diagnosis, antifungal treatment, mainly an echinocandin, was given, with the exception of three patients who died because of the severity of their acute illness before blood culture results were available. After the susceptibility results became available, the initial treatment could be modified by the attending intensivists.

Characteristics of COVID-19 patients who developed candidemia were compared with those of two historical candidemia cohorts from our ICU before the COVID-19 pandemic—in particular, an earlier cohort including all ICU patients who developed candidemia from 2005 to 2008 (*n* = 66) and a later one comprising all ICU patients who developed candidemia from 2012 to 2015 (*n* = 77).

### 2.2. Definitions

ICU-acquired candidemia was defined as the presence of at least one positive blood culture for any *Candida* species in the blood specimen collected more than 48 h after ICU admission. Blood cultures were performed in the presence of signs and symptoms of sepsis or when infection was suspected on clinical rounds. The onset of candidemia was defined as the specimen collection date for the positive *Candida* blood culture.

### 2.3. Species Identification and Antifungal Susceptibility Testing

The BD Bactec (Becton Dickinson, Sparks, MD, USA) automated blood culture system was used for monitoring blood culture bottles. Fungal isolates were identified at species level by using the VitekMS (BioMeriéux, Marcy l’Etoile, France) device and MALDI-TOF MS method. Antifungal susceptibility was evaluated with the Vitek2 (BioMeriéux, Marcy l’Etoile, France) automated system. The phenotypic susceptibility profile for each fungal isolate was interpreted according to the EUCAST standard (European Committee on Antimicrobial Susceptibility Testing Breakpoint tables for interpretation of MICs for antifungal agents, Version 10.0, valid from 4 February 2020). In addition, for the period of March 2020 to October 2021, directly from the signal-positive blood culture vials with yeasts in Gram staining, a multiplex syndromic approach was applied, namely the FilmArray Blood Culture Identification 2 panel (BCID2 assay, BioMeriéux, Marcy l’Etoile, France), for the early detection of the emerging species *Candida auris*.

### 2.4. Statistical Analysis

Statistical data analysis was performed using the R software, Version 4.1.1 (R Foundation for Statistics, Vienna, Austria). Data are described as mean ± SD or median and interquartile range (IQR) in case of variables with non-normal distribution, and as number and percentage (%) in case of categorical variables. In order to compare the distributions of numerical variables between two groups of patients, we used the two-sample *t*-test, or, alternatively, the Mann–Whitney U test in case of variables with non-normal distribution, whereas associations between qualitative factors were appropriately investigated via the chi-squared (X^2^) statistic or Fisher’s exact test. Incidence between the various cohorts was also compared via the statistical test of proportions. Univariate and multivariate binary logistic regression models were built for the determination of risk factors for bloodstream infection with *albicans* versus non-*albicans Candida* species and for potentially fluconazole-resistant species, reporting odds ratios (OR) and corresponding 95% confidence intervals (CI) in relation to the model covariates. The level of statistical significance was set at 0.05.

## 3. Results

### 3.1. COVID-19 Candidemia Cohort

In the 18-month study period during the pandemic, among 600 patients who were admitted to the ICU due to COVID-19, 62 patients developed candidemia during the ICU stay, accounting for an incidence of 10.2%. The median [IQR] age of the patients with candidemia was 69 [15.8] years, and 72.4% were males. Admission APACHE II and SOFA scores were 15 [7] and 8 [3], respectively.

The median [IQR] time between hospital and ICU admission and positive *Candida* culture was 28.5 [19.5] days and 22 [18.2] days, respectively. Non-*albicans Candida* species predominated (in 47 out of 62 patients, 77%). Among the non-*albicans* species, the most frequently isolated was *C*. *parapsilosis* (31 patients, 50%), followed by *C. auris* (9 patients, 14%) and *C. glabrata* (6 patients, 9.7%).

### 3.2. Comparison between COVID-19 Candidemia Cohort and the Pre-COVID-19 Cohorts

Baseline characteristics of the ICU patients with candidemia development, stratified according to the time period of ICU admission, are presented in Table 1. Compared with patients without COVID-19, patients with COVID-19 were older and had lower illness severity as expressed by APACHE II and SOFA scores on ICU admission. However, on candidemia day, they were more likely to present circulatory shock and they had a higher SOFA score. As a result, the Delta SOFA score was significantly higher in COVID-19 patients than in the non-COVID-19 ones (3 (6) versus 0 (4) and −1 (3.5), respectively, *p* < 0.001). As expected, patients with COVID-19 were less likely to have a surgical diagnosis on ICU admission. Patients with and without COVID-19 had similar hospital and ICU LOS before candidemia development. While the incidence of candidemia did not change significantly between 2005–2008 and 2012–2015, a significant increase was observed in the COVID-19 cohort compared to the two pre-pandemic cohorts (10.2% (62/600) versus 3.8% (66/1737) and 4.2% (77/1833), respectively, *p* < 0.001). All-cause ICU mortality was 47.8% for *C. albicans* and 59% for non-*albicans Candida.* There were no differences in mortality rates among the three periods; see Table 1.

### 3.3. Candida Species Distribution and Fluconazole Resistance

The distribution of *Candida* species and the antifungal susceptibility during the three study periods are shown in Table 2. Non-*albicans Candida* species predominated in all cohorts, with *C. parapsilosis* being the most commonly isolated. Considerable differences in *Candida* species distributions were observed over the years. In particular, a gradual decrease in the incidence of *C. albicans* was observed in the COVID-19 pandemic cohort (from 39.4% in 2005–2008 and 37.7% in 2012–2015 to 24.2% in COVID-19 cohort), accompanied by a corresponding increase in non-*albicans Candida* species, including the emergence of *C. auris*; see Figure 1.

During the COVID-19 period, fluconazole resistance occurred in 30 (48.4%) candidemia cases: 2/15 in *C. albicans*, 17/31 in *C. parapsilosis*, 3/6 in *C. glabrata*, 9/9 in *C. auris*; see Table 2. Fluconazole resistance considerably increased over the three time periods, from 31.8% in 2005–2008, to 37.7% in 2012–2015, and to 48.4% in the COVID-19 period, *p* = 0.098; see Table 2 and Figure 1.

Regarding the susceptibility tests for other antifungal agents, we did not observe resistance of the aforementioned *Candida* species to amphοtericin B, echinocandins and voriconazole.

### 3.4. Factors Associated with Non-Albicans Candidemia

Regarding the entire cohort of patients who developed candidemia during the three time periods (*n* = 205), factors associated with non-*albicans Candida* species, according to univariate and multivariate models, are shown in Table 3.

Multivariate logistic regression analysis revealed that an increased SOFA score on candidemia day (compared to that on ICU admission) was independently associated with candidemia due to *Candida albicans*, whereas the presence of shock on candidemia day was independently associated with candidemia due to non*-albicans Candida* species; see Table 3.

### 3.5. Factors Associated with Fluconazole Resistance

Resistance to fluconazole was significantly associated with non-*albicans Candida* species (54.8% versus 8.6%, in non-*albicans Candida* species and *C. albicans,* respectively, *p* < 0.001); see Figure 2. Factors associated with fluconazole resistance are shown in Table 4. Compared to patients who developed candidemia due to fluconazole-susceptible *Candida* species, patients with fluconazole-resistant strains had longer hospital and ICU LOS before the onset of candidemia (33 (27) versus 23 (22.8) days, *p* = 0.03, and 26 (22.5) versus 16 (21) days, *p* = 0.003, respectively). Multivariate analysis showed that prolonged ICU LOS before candidemia onset was significantly associated with the development of candidemia due to fluconazole-resistant *Candida* species (OR 1.03, CI: 1.01–1.06, *p* = 0.003), whereas the presence of shock at candidemia onset was independently associated with candidemia due to fluconazole-susceptible *Candida* species (OR 0.23, CI: 0.07–0.64, *p* = 0.006); see Table 4.

## 4. Discussion

This study investigated the incidence and epidemiology of candidemia in patients admitted to the ICU due to COVID-19, compared to two previous non-COVID-19 ICU candidemia cohorts. The main findings are the following: (i) candidemia incidence was 10%, more than two-fold higher compared to the pre-pandemic era; (ii) there was an epidemiological shift to non-*albicans Candida* species from 60.6% to 75.8% with a predominance of *C. parapsilosis* and (iii) there was a considerable increase in the rate of fluconazole resistance from 31.8% to 48.4%. In addition, for the whole cohort of patients with candidemia, fluconazole resistance was independently associated with ICU LOS before candidemia onset, whereas fluconazole susceptibility was independently associated with the presence of shock at candidemia onset.

The increase in the incidence of candidemia shown in our study during the ongoing pandemic is striking, though consistent with initial findings from our ICU in the first pandemic wave [8], as well as with findings of other institutions in different geographic regions [6,11,14,16,26,27,28]. In particular, in studies comparing patients with and without COVID-19, a two-fold increase in the incidence of candidemia in COVID-19 compared to non-COVID-19 patients was observed in two ICUs in India [29], whereas a nearly five-fold increase has been reported in Brazil [16], and a 10-fold rise in another report [27]. Similarly, in another case series from Italy, a higher incidence of candidemia in COVID-19 patients compared with a historical control has been reported [11], though, in the latter two studies, information about patients’ hospital location (i.e., ICU or ward) was not reported.

There was no difference in the incidence of candidemia in our ICU between the two pre-pandemic periods. This is in accordance with nationwide data from Germany showing that there was no increase in the ICU-acquired candidemia incidence during the period from 2006 to 2011 [30]. However, an increasing incidence of candidemia has been reported in internal medicine wards in our country, possibly associated with the financial crisis [20]; the present study shows that COVID-19 has further accelerated the phenomenon.

In fact, the above findings are not surprising since critically ill patients with COVID-19 have similar risk factors for candidemia development with the other, non-neutropenic ICU patients and they also received corticosteroid treatment, as recommended after the first pandemic wave [31], which might have been an additional risk factor, as already commented elsewhere [27]. Furthermore, over-occupancy of the ICU, along with the higher workload of healthcare workers and the subsequent relaxation in compliance with the infection control measures, might be additional causes [28].

The increased incidence of non*-albicans*
*Candida* species detected in our study is consistent with comparable data previously reported from our ICU [32] and elsewhere [19], as well as with recent data demonstrating an increasing incidence of candidemia in a nationwide study from Greece, with a species shift towards *C. parapsilosis* [33]. The increased incidence of non*-albicans*
*Candida* species is in accordance with an epidemiological shift across the globe, including the emergence of non*-albicans Candida* species. Indicatively, in a recent study [27], non-*albicans Candida* collectively constituted the majority of isolates in candidemic patients, considering non-COVID-19 and COVID-19 cases together. Similarly, in another study from India during the current pandemic [30], 64% of candidemia cases were due to non-*albicans Candida* species. On the contrary, *C. albicans* remains the predominant pathogen of candidemia in COVID-19 patients in Europe [11,13,34,35,36], as well as in the pre-pandemic era, according to German data for candidemia in the ICUs [31].

Introduced in the early 1990s, fluconazole is an often-preferred treatment for many systemic *Candida* infections as it is inexpensive and exhibits limited toxicity; it is implicated, however, in the subsequent resistance acquisition due to its extensive use over the years [19,37].

According to our results, concomitantly to the increase in non-*albicans Candida* species, a worrisome increase in the rate of fluconazole resistance was observed, from around 32% in the pre-COVID-19 era to 48% in the COVID-19 period. Although not surprising, since fluconazole resistance is closely associated with non-*albicans Candida* species, such an increase in fluconazole-resistant *Candida* species is of major concern. Notably, among the various isolated species, *C. parapsilosis* presented the highest proportion of resistance of around 50% across all three time periods, with the exception of *C. auris*, which has expected fluconazole resistance as a potential multidrug-resistant yeast. All-cause mortality did not differ significantly among the three study periods.

In our analysis, at candidemia onset, the SOFA score was significantly higher and the presence of shock was more frequent in patients with COVID-19 compared to those in the pre-pandemic periods, indicating an excess severity of the multi-organ dysfunction in those patients, though without a significant increase in mortality.

Interestingly, considering non-COVID-19 and COVID-19 cases together, the results of multivariate analyses revealed that the presence of shock at candidemia onset was independently associated with the isolation of *Candida albicans* and with fluconazole-susceptible species. This is a novel finding, possibly suggesting a more virulent capacity of the *Candida albicans* compared to non*-albicans* species. Although this finding is consistent with recent experimental data [38], it deserves further research.

Certain limitations of the present study should be pointed out. The first is the absence of a contemporaneous group to the COVID-19 pandemic cohort, i.e., critically ill patients admitted to the non-COVID-19 ICUs during the current pandemic; thus, the comparisons have been made with pre-pandemic cohorts. In fact, such a design was not feasible since the majority of our ICU beds were dedicated entirely to the admission of COVID-19 patients. However, thanks to the availability of data from the two historical non-COVID-19 cohorts, the trends in the incidence and epidemiology of candidemia in our ICU have been shown. Secondly, consumption of antifungal drugs, namely fluconazole, at the individual patient level before the onset of candidemia has not been recorded. However, in our ICU, there was no routine prophylactic use of antifungals, as has already been reported [31]; therefore, pre-exposure to fluconazole is less likely to have influenced these results.

Finally, comparisons between patients with and without candidemia during the study periods were not available, since only patients who developed candidemia have been included in the analysis. However, despite the above limitations, the present study highlights the importance for the critical care teams to be aware of the increased incidence of candidemia and of fluconazole resistance in COVID-19 ICU patients, in order to recognize cases early and treat them accordingly, as well as the urgent need to integrate antimicrobial stewardship activities in the pandemic response.

## 5. Conclusions

In summary, the present study provides temporal trends for candidemia in an ICU setting. The incidence of candidemia was significantly increased during the COVID-19 pandemic compared to previous non-COVID-19 periods. Additionally, a substantial increase in the incidence of non-*albicans Candida* and in fluconazole-resistant species was observed during the COVID-19 era. Prolonged ICU LOS was associated with fluconazole-resistant and the presence of shock with flunonazole-susceptible species. Further study is needed to clarify the reasons for the increased incidence of candidemia and of fluconazole resistance in the COVID-19 ICU patients. Meanwhile, the present findings underscore the urgent need for increased awareness, as well as for the implementation of antimicrobial and antifungal stewardship programs in order to diminish the incidence of candidemia and fluconazole resistance rates.

## Figures and Tables

**Figure 1 antibiotics-11-00771-f001:**
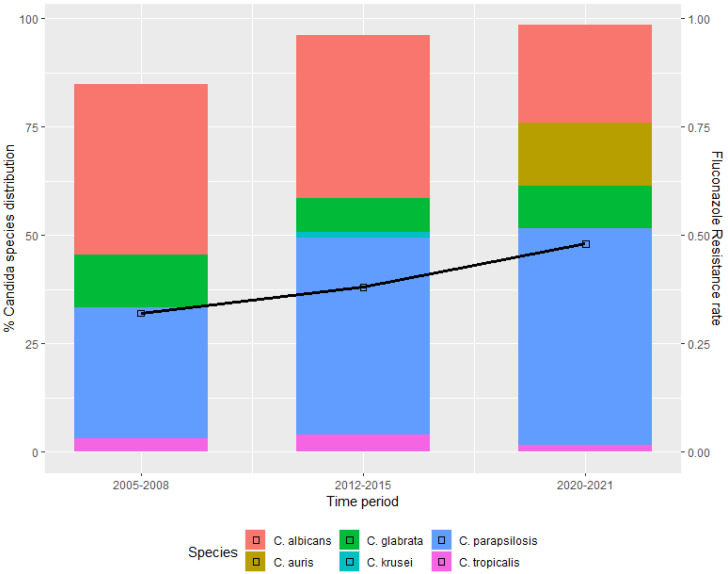
Species distribution of *Candida* bloodstream isolates and fluconazole resistance before and during the COVID-19 pandemic era.

**Figure 2 antibiotics-11-00771-f002:**
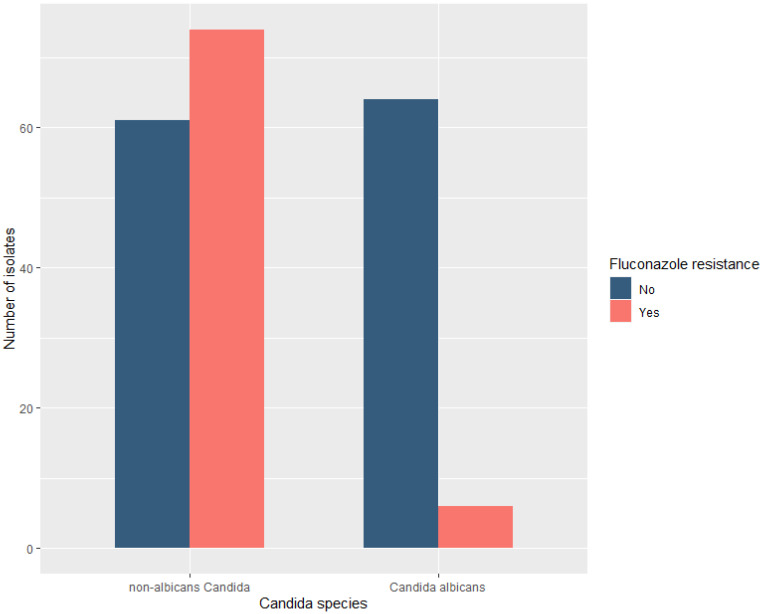
Fluconazole resistance in all (*n* = 205) bloodstream-isolated *Candida albicans* and non-*albicans Candida* species.

**Table 1 antibiotics-11-00771-t001:** Characteristics of ICU patients with candidemia in the three candidemia cohorts.

Variables	Pre-COVID-19 Cohorts		COVID-19 Cohort	*p*-Value
	2005–2008 *n* = 66	2012–2015 *n* = 77	2019–2021 *n* = 62	
**Age, median (IQR)**	67 (21)	63 (31)	69 (15.8)	**0.001**
males, *n* (%)	45 (68.1)	46 (59.7)	45 (72.4)	0.27
APACHE II score on ICU admission, median (IQR)	19 (8.8)	20 (10)	15 (7)	<**0.001**
SOFA score on ICU admission, median (IQR)	9 (4)	10 (5)	8 (3)	**0.001**
SOFA score on candidemia day, median (IQR)	8.5 (6)	7 (5)	11 (6)	**0.001**
Delta SOFA score, median (IQR)	0 (4)	−1 (3.5)	3 (6)	<**0.001**
ICU admission diagnosis Medical, *n* (%) Surgical, *n* (%)	22 (33) 44 (66.7)	40 (53.3) 35 (46.7)	62 (100) 0 (0)	<**0.001**
Co-morbidities Diabetes mellitus, *n* (%) Current malignancy, *n* (%)	5 (7.6) 6 (9.1)	3 (3.9) 5 (6.5)	16 (25.8) 5 (8.1)	<**0.001** 0.99
Hospital stay before candidemia onset, days, median (IQR)	24 (18)	30 (41.8)	28.5 (19.5)	0.15
ICU stay before candidemia onset, days, median (IQR)	15.5 (19.8)	25 (34.5)	22 (18.2)	0.69
ICU length of stay, days, median (IQR)	35.5 (34.5)	49 (51)	34.5 (39.8)	**0.029**
Presence of shock on candidemia day, *n* (%)	34 (52.3)	30 (46.1)	45 (75)	**0.001**
Incidence of candidemia, (%)	3.8	4.2	10.3	<**0.001**
Mortality, *n* (%)	42 (63.6)	35 (46.7)	35 (56.5)	0.92

ICU, intensive care unit; IQR, interquartile range; APACHE, Acute Physiology and Chronic Health Evaluation; SOFA, Sequential Organ Failure Assessment; Delta SOFA, SOFA score on candidemia day minus SOFA score on ICU admission.

**Table 2 antibiotics-11-00771-t002:** *Candida* species and fluconazole resistance in the three candidemia cohorts.

	Pre-COVID-19 Cohorts		COVID-19 Cohort	*p*
	2005–2008 *n* = 66	2012–2015 *n* = 77	2020–2021 *n* = 62	
** *Candida* ** **species**				
*C. albicans, n* (%)	26 (39.4)	29 (37.7)	15 (24.2)	0.069
non*-albicans Candida, n* (%)	40 (60.6)	48 (62.3)	47 (75.8)	0.069
* C. parapsilosis*	28 (70)	36 (75)	31 (66)	0.16
* C. glabrata*	8 (20)	6 (12.5)	5 (10.6)	1
* C. tropicalis*	2 (5)	3 (6.3)	1 (2)	0.77
* C. krusei*	0 (0)	1 (2)	0 (0)	1
* C. auris*	0 (0)	0 (0)	9 (19)	**<0.001**
other *Candida* species	2 (5)	2		1
**Fluconazole resistance**				
Fluconazole-resistant, *n* (%)	21 (31.8)	29 (37.7)	30/62 (48.4)	0.098
* C. albicans*	4/26 (15.4)	1/29 (3.4)	2/15 (13.3)	
* C. parapsilosis*	10/28 (35.7)	20/36 (55.6)	17/31 (48.6)	
* C. glabrata*	7/8 (87.5)	2/6 (33.3)	2/5 (40)	
* C. tropicalis*	0/2 (0)	0/3 (0)	0/1 (0)	
* C. krusei*	NA	1/1(100)	NA	
* C. auris*	NA	NA	9/9 (100)	

NA: non-applicable.

**Table 3 antibiotics-11-00771-t003:** Factors associated with candidemia development due to *Candida albicans* versus non-*albicans Candida* species in the overall study population: univariate and multivariate models. OR (95% CI) takes non-*albicans Candida* as the reference group.

	Patients with Candidemia, *n* = 205			
	*Candida albicans* Species, *n* = 70	Non-*albicans Candida* Species, *n* = 135	OR (95% CI)	*p*-Value
Univariate analysis				
Age (years) ‡	63.0 (22.0)	67.0 (21.0)	0.98 (0.97–1.01) ^b^	0.19
Gender (Female), *n* (%)	24 (34.3%)	45 (33.3%)	1.04 (0.56–1.91)	0.89
ICU stay before candidaemia onset, days ‡	15.0 (19.2)	23.0 (24.5)	0.98 (0.9–1.00) ^c^	0.08
Hospital stay before candidaemia onset, days ‡	23 (25)	29.5 (24)	0.99 (0.98–1.01) ^c^	0.17
Delta SOFA	−0.32 ± 4.07	1.10 ± 4.15	0.91 (0.84–0.98) ^d^	**0.03**
ICU length of stay, days ‡	39.0 (36.5)	38.0 (37.0)	0.99 (0.99–1.01) ^c^	0.90
Diagnosis (surgical), *n* (%)	30 (43.5%)	49 (36.6%)	1.33 (0.73–2.41)	0.33
Presence of shock on candidemia day, *n* (%)	36 (54.5%)	73 (58.9%)	0.83 (0.45–1.53)	0.56
COVID-19	15 (24.2%)	47 (75.8%)	0.51 (0.25–0.98)	**0.049**
Multivariate analysis ^a^				
ICU stay before candidemia onset, days			0.97 (0.95–1.00) ^c^	0.08
Delta SOFA			0.74 (0.60–0.89) ^d^	**0.002**
Presence of shock on candidemia day			6.89 (2.2–25.0)	**0.001**

‡: Median (IQR) for skewed parameters; OR: odds ratio; CI: confidence interval. ^a^ Significant results adjusted for other variables in the model; ^b^ per each year increase; ^c^ per each day increase; ^d^ per each unit increase; ICU, intensive care unit; SOFA, Sequential Organ Failure Assessment; Delta SOFA, SOFA score on candidemia day minus SOFA score on ICU admission.

**Table 4 antibiotics-11-00771-t004:** Factors associated with candidemia development due to fluconazole-resistant *Candida* species in the overall patient population: univariate and multivariate models. OR (95% CI) takes fluconazole-susceptible as the reference group.

	Patients with Candidemia, *n* = 205			
	Fluconazole-Resistant Species, *n* = 80	Fluconazole-Susceptible Species, *n* = 125	OR (95% CI)	*p*-Value
Univariate analysis				
Age (years) ‡	67.0 (20.8)	65.5 (22.5)	1.01 (0.98–1.03) ^b^	0.22
Gender (Female), *n* (%)	29 (36.2%)	40 (32.0%)	1.20 (0.66–2.17)	0.53
ICU stay before candidemia onset, days ‡	26.0 (22.5)	16 (21.0)	**1.02 (1.01–1.04) ^c^**	**0.003**
Hospital stay before candidemia onset, days ‡	33.0 (27.0)	23.0 (22.8)	**1.01 (1.00–1.03) ^c^**	**0.03**
Delta SOFA	0.49 ± 4.64	0.68 ± 3.87	0.98 (0.91–1.06) ^d^	0.76
ICU length of stay, days ‡	43.0 (48.0)	36.0 (37.0)	1.01 (0.99–1.02) ^c^	0.09
Diagnosis (surgical), *n* (%)	30 (38.0%)	49 (39.5%)	0.93 (0.52–1.66)	0.82
Presence of shock on candidemia day, *n* (%)	37 (52.9%)	72 (60.0%)	0.74 (0.41–1.35)	0.33
COVID-19	30 (48.4%)	32 (51.6%)	1.74 (0.95–3.20)	0.07
Multivariate analysis ^a^				
ICU stay before candidemia onset			1.03 (1.01–1.06)	**0.003**
Presence of shock on candidemia day			0.23 (0.07–0.64)	**0.006**

‡: Median (IQR) for skewed parameters; OR: odds ratio; CI: confidence interval. ^a^ Significant results adjusted for other variables in the model; ^b^ per each year increase; ^c^ per each day increase; ^d^ compared to Day 0; ICU, intensive care unit; SOFA, Sequential Organ Failure Assessment; Delta SOFA, SOFA score on candidemia day minus SOFA score on ICU admission.

## Data Availability

Data supporting the results can be provided from the corresponding author on request.

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
