# Peer review of "Epidemiology of Candidemia and Fluconazole Resistance in an ICU before and during the COVID-19 Pandemic Era"

_antibiotics, 2022, doi:10.3390/antibiotics11060771_

Round 1

Reviewer 1 Report

Very interesting paper and well wrote paper. The topic is very relevant also because Candidemia in ICU are difficult to treat due to fragile patients.

Below my suggestions

  1. Introduction: updata data on SARS CoV2 wirdwilde and introduce better the role of AMR and Candidemia during COVID 19 pandemic (see and cite Segala FV. Impact of SARS-CoV-2 Epidemic on Antimicrobial Resistance: A Literature Review. Viruses. 2021 Oct 20;13(11):2110. doi: 10.3390/v13112110.)
  2. Methods and result: very well presented
  3. Discussion:suggest some infection prevention control that came fro  your interesting data. How suggest to prevent candidemia?
  4. Also discuss the role of medical education in Candidemia treatment and early symptom and diagnosis
  5. Discuss also the risk factor for Candidemia in your patiens and in other studies
  6. During Candidemia the patient remain also in antibiotic therapy or stop antibiotic therapy?
  7. Is there some concomitant bacteria infection ?

Author Response

Very interesting paper and well wrote paper. The topic is very relevant also because Candidemia in ICU are difficult to treat due to fragile patients.

Below my suggestions

  1. Introduction: updata data on SARS CoV2 wirdwilde and introduce better the role of AMR and Candidemia during COVID 19 pandemic (see and cite Segala FV. Impact of SARS-CoV-2 Epidemic on Antimicrobial Resistance: A Literature Review. Viruses. 2021 Oct 20;13(11):2110. doi: 10.3390/v13112110.)

Answer

Data on SARS CoV2 have been updated (Ref 1 and 2). The role of AMR during the pandemic and the suggested article have been added ( Intoduction, lines 45-47, and new Reference No 10).

  1. Methods and result: very well presented

We would like to thank you for your comment

  1. Discussion:suggest some infection prevention control that came fro  your interesting data. How suggest to prevent candidemia?

According to your suggestion, we comment on this: lines 303-307:

“However, despite the above limitations, the present study highlights the importance for the critical care teams to be aware of the increased incidence of candidemia and of fluconazole resistance in COVID-19 ICU patients, in order to early recognize and treat accordingly, as well as the urgent need to integrate antimicrobial stewardship activities in the pandemic response.”

  1. Also discuss the role of medical education in Candidemia treatment and early symptom and diagnosis

We comment on this in the same paragraph as above

  1. Discuss also the risk factor for Candidemia in your patiens and in other studies

Answer

We discuss the possible risk factors for candidemia in COVID-19 era in the Discussion part, lines 247-253. We have also reported risk factors for candidemia in our ICU previously (Mycoses 2011:54,154, Ref 32). Unfortunately, it is not possible to provide specific risk factors in the present patient population because patients without candidemia have not been included in this cohort, and therefore, in the analysis. However, thanks to your comment, a relevant limitation has been added (Limitations, last paragraph lines 301-307).

  1. During Candidemia the patient remain also in antibiotic therapy or stop antibiotic therapy?

Answer

Usually, there was a long ICU stay before candidemia onset (15.5 to 25 days, Table 1). Meanwhile, a bacterial infection had already occurred (as shown in a previous article from our ICU (J Hosp Infect 2021; 107:95, Ref No. 9); so, almost all candidemic patients were already under antimicrobial treatment.  

  1. Is there some concomitant bacteria infection ?

Answer

A polymicrobial population (a Candida strain along with a bacterial one), simultaneously grown by the same blood culture occurred in just one patient only.

Concomitant bacterial infections (blood stream infection or ventilator associated pneumonia)  were usually diagnosed prior to candidemia onset in the majority of patients (please see our answer to your previous comment).

Reviewer 2 Report

Dear authors

The manuscript is well structured, needs adjustment.

I believe that editing and rewriting  same part of the text is necessary to make the manuscript easily readable, but this effort would be well worth it as the authors have included a great deal of information.

However, I have following comments that should be addressed:

In the introduction part first row need to be rewrite.

The article need to be double check by an native English speaker.

The conclusion need to focus on clinical improvement of patients treatment, and to introduce in clinical use.  The conclusion part need more explanation and clear points. 

Please streamline 3 short conclusion!

  • Please recheck the References order
  • Double Check abreviation!
  • Table 1 is hard to read!

Author Response

The manuscript is well structured, needs adjustment.

I believe that editing and rewriting  same part of the text is necessary to make the manuscript easily readable, but this effort would be well worth it as the authors have included a great deal of information.

However, I have following comments that should be addressed:

In the introduction part first row need to be rewrite.

Answer

The first sentence has been rewritten, according to your suggestion.

The article need to be double check by an native English speaker.

Answer

English editing has been done

The conclusion need to focus on clinical improvement of patients treatment, and to introduce in clinical use.  The conclusion part need more explanation and clear points. 

Please streamline 3 short conclusion!

Answer

We have improved the conclusion part according to your suggestion

  • Please recheck the References order

Reference order has been checked

  • Double Check abreviation!

Abbreviations have been checked

  • Table 1 is hard to read!

We tried, but it is not possible to further simplify the Table 1, so, we leave it on the Editor’s discretion